# A Comparative Study of Functional Connectivity Measures for Brain Network Analysis in the Context of AD Detection with EEG

**DOI:** 10.3390/e23111553

**Published:** 2021-11-22

**Authors:** Majd Abazid, Nesma Houmani, Jerome Boudy, Bernadette Dorizzi, Jean Mariani, Kiyoka Kinugawa

**Affiliations:** 1SAMOVAR, Télécom SudParis, Institut Polytechnique de Paris, 9 Rue Charles Fourier, F-91011 Évry, France; majd.abazid@telecom-sudparis.eu (M.A.); jerome.boudy@telecom-sudparis.eu (J.B.); bernadette.dorizzi@telecom-sudparis.eu (B.D.); 2Sorbonne Université, CNRS, UMR 8256 Biological Adaptation and Aging, F-75005 Paris, France; jean.mariani@sorbonne-universite.fr (J.M.); kiyoka.kinugawa@aphp.fr (K.K.); 3Sorbonne Université, UFR Médecine, F-75013 Paris, France; 4Assistance Publique—Hôpitaux de Paris (AP-HP), DHU FAST, Functional Explorations and Sleep Investigation Unit for the Older Patients, Charles Foix Hospital, F-94200 Ivry-sur-Seine, France

**Keywords:** EEG signals, graph theory, brain network, epoch-based entropy, phase lag index, coherence, mutual information, mild cognitive impairment, subjective cognitive impairment, AD detection

## Abstract

This work addresses brain network analysis considering different clinical severity stages of cognitive dysfunction, based on resting-state electroencephalography (EEG). We use a cohort acquired in real-life clinical conditions, which contains EEG data of subjective cognitive impairment (SCI) patients, mild cognitive impairment (MCI) patients, and Alzheimer’s disease (AD) patients. We propose to exploit an epoch-based entropy measure to quantify the connectivity links in the networks. This entropy measure relies on a refined statistical modeling of EEG signals with Hidden Markov Models, which allow a better estimation of the spatiotemporal characteristics of EEG signals. We also propose to conduct a comparative study by considering three other measures largely used in the literature: phase lag index, coherence, and mutual information. We calculated such measures at different frequency bands and computed different local graph parameters considering different proportional threshold values for a binary network analysis. After applying a feature selection procedure to determine the most relevant features for classification performance with a linear Support Vector Machine algorithm, our study demonstrates the effectiveness of the statistical entropy measure for analyzing the brain network in patients with different stages of cognitive dysfunction.

## 1. Introduction

Electroencephalography (EEG) is considered as a convenient acquisition technology for brain activity analysis in the clinical context. Indeed, it is appropriate for cognitively and physically disabled patients, as well as for serial tests in the absence of objective cognitive decline [1]. Moreover, EEG has the advantage of being a non-invasive, cost-effective, widely available and mobile technology. Additionally, it is characterized by a high temporal resolution (i.e., milliseconds), which is essential for the study of fast brain functional dynamics.

The literature has largely highlighted that EEG coupled with appropriate signal processing techniques can provide precious information on normal and impaired brain networks [2]. There is a rich literature addressing the use of EEG to investigate brain activity alterations due to neurodegenerative diseases (NDD), especially Alzheimer’s disease (AD), which is the most prevalent form of NDD [3]. AD is a chronic and irreversible disease that produces a progressive cognitive decline. We observe a growing interest in the earlier stages of the disease since curative treatments are still lacking. The preclinical stage of AD is asymptomatic, but the brain lesions due to AD are present. At this phase, the term of subjective cognitive impairment (SCI) is defined and refers to when a patient presents a self-reported experience of persistent deterioration in cognitive function, which cannot be verified by standard tests [4,5,6]. In the mild cognitive impairment (MCI) stage, patients show measurable memory impairments but their functional capacity is maintained [5,7,8]. SCI and MCI patients are at high risk of developing AD [5,6]. Diagnosing AD with EEG at the early stage is still a challenge due, on one hand, to the fact that early symptoms are dismissed as the normal effects of ageing and, on the other hand, to the complex nature of EEG signals, characterized as nonstationary, nonlinear and multidimensional biosignals.

Numerous studies in the literature have revealed changes in EEG signals recorded in eyes-closed resting-state condition at the early stage of AD. It is largely known that AD induces a reduction in the complexity of EEG signals and an alteration in EEG functional connectivity. These changes in EEG signals have been exploited as discriminative features for AD diagnosis. Various methods have been used to quantify the complexity of EEG signals, such as the correlation dimensions and the first positive Lyapunov exponent [9,10,11,12,13,14]. However, these two measures involve the reconstruction of a phase space trajectory, which requires a high computation cost. Other methods inherited to information theory, entropy-based measures in particular, have also been proposed, such as sample entropy [15], Tsallis entropy [16], approximate entropy [17,18], and multiscale entropy [19]. These methods associate the complexity of a signal to its irregularity and unpredictability.

Usually, these measures were applied with two main drawbacks. First, they were applied on EEG signals without considering the nonstationarity and nonlinearity properties of EEG signals. The assumption of stationarity and linearity are generally not appropriate for physiological data. In [20], the authors pointed out that nonstationarity is an intrinsic property of physiological data, even without external stimulus. In [21], the authors claimed that the human brain is a complex system generating nonstationary and nonlinear signals. Nonstationarity means that the statistical properties of the signal varies with time. The authors suggested that the observed nonstationarity in EEG signals reflects a switching of the inherent metastable states of neural assemblies during brain functioning. In [22], the authors claimed that the EEG signal can be modeled as a sequence of quasi-stationary epochs separated by sudden transitions. In [23], the authors indicated that EEG signals are characterized in terms of metastability, which refers to the ability of the brain to move from one stable state to another, remaining for an extended time period. In [24], EEG is described as a piecewise stationary process, segmented into stationary epochs with different probabilistic characteristics. Additional studies [25,26,27] identified quasi-stationary states in EEG, referred to as “microstates”, reflecting coherent neural dynamics.

Secondly, such measures did not exploit the EEG signal as a multivariate time series. Actually, the predominant approach in the literature consists of extracting information from EEG signals by averaging them over channels. The EEG signal being a multidimensional signal recorded by a multiplicity of electrodes (channels), it is of high interest to take advantage of its spatiotemporal nature using techniques that can catch the inter-channel relations. In that sense, alternative methods were used for assessing the inter-channel relations, such as mutual information [28,29], coherence [29,30,31], Granger causality [29,32] and phase lag index [33,34]. Nevertheless, these measures quantify only the information transmission between different channels, without considering the temporal characteristics of EEG signals.

More recently, graph theory has gained considerable ground to study normal and impaired brain networks [35,36,37,38,39,40,41,42,43,44,45,46,47,48]. By modeling a brain network as a graph consisting of nodes (electrodes) linked by edges representing the connectivity between cortical nodes, it is possible to carry out a topological investigation of the brain functional organization. Many studies have shown that the topology of the brain network is altered in AD and MCI patients [38,39,41,42,43,44,45,46,47,48]. Some studies have pointed out that the AD group deviates from the optimal small-world topology exhibiting a more random one compared to control subjects [41,42]. Concerning the other topological parameters, conflicting results emerge in some respects [41,42,43,44,45,46,48]. This is mainly due to the use of sparse data sets with different characteristics as well as methodological differences. Indeed, the majority of works exploit databases with different characteristics, and which are subject to experimental constraints that do not correspond to the reality on the ground. They can include strict patient’s inclusion and/or exclusion criteria, and consider normal healthy subjects as controls. Moreover, most studies use binary graph networks that necessitate the application of an optimal threshold on functional connectivity matrices. Usually, such a threshold value is chosen empirically. This factor affects directly the resulting network. Moreover, different metrics are used to quantify the connectivity links in brain networks. Intuitively, these classical metrics may reflect different processes leading to different brain network topologies. The majority of studies encouraged the use of a specific metric, without comparing it to others on the same database. Additionally, it is important to notice that all graph-based studies in AD express links in graph networks using only the degree of signal synchronization between different electrodes, without taking into account the complete spatiotemporal alterations due to AD, namely the reduction in both complexity and inter-channel connectivity.

In this work, we propose to use an entropy-based entropy measure, called “epoch-based entropy” (EpEn), already introduced and published in [49,50,51,52]. This measure relies on the statistical modeling of EEG signals with a Hidden Markov Model (HMM), which considers the propriety of nonlinearity, nonstationarity and multidimensionality of EEG signals. In [49,50,51,52], we demonstrated that EpEn leads to a better characterization of the dynamic system underlying the observed EEG signals for AD detection since HMM allows a refined statistical modeling of the spatiotemporal changes in EEG time series. More specifically, EpEn quantifies the information content of EEG signals, both at the time and spatial levels, using local density estimation by an HMM, on inter-channel quasi-stationary epochs (segments).

The present study addresses AD detection with different functional connectivity measures using a database containing EEG signals acquired in real clinical conditions. This database contains EEG data from SCI patients, mild to moderate AD patients and MCI patients. The objective of the study is to extend the use of EpEn to brain network assessment and demonstrate its effectiveness with different graph parameters. We also conduct a comparative study by investigating the classification performance of EpEn as well as three additional connectivity measures, namely phase lag index, coherence and mutual information, when exploited to quantify the connectivity links in brain networks.

## 2. Materials and Methods

### 2.1. Methodology

The present study was conducted on a cohort containing EEG data of three populations, SCI, MCI and mild to moderate AD patients, recorded in real-life clinical conditions. Each person in the database had 30 EEG recordings captured by the used 30 electrodes.

In order to discriminate between SCI, MCI and AD, we first computed the functional connectivity in the three groups using epoch-based entropy (EpEn) measure, and that for each frequency band (1–4 Hz delta, 4–8 Hz theta, 8–12 Hz alpha, 12–30 Hz beta). The connectivity measure was computed between all pairs of the 30 electrodes for each person. Then, we applied a thresholding on the obtained real-valued connectivity matrices to generate their binary form keeping only the highest connectivity values. We considered 9 threshold values from 10% to 90% with steps of 10%. Therefore, for each threshold value (among the nine), a person had four binary matrices (in the four frequency bands), on which we computed a local graph parameter to characterize the topology of the binary network. A vector of 30 values was thus obtained for each matrix, where each element is the graph parameter value of a node (electrode).

After that, a feature selection procedure was applied to identify the most relevant nodes in each frequency band for a given threshold. To do that, we used the Orthogonal Forward Regression (OFR) algorithm and the random probe technique [53,54,55]. Then, by merging the selected features of the four frequency bands, we applied an additional feature selection procedure to combine the extracted information at different frequency bands. At the end, the selected features represented the graph parameter value for the selected nodes and the selected frequency bands.

Finally, such selected combination of features was given as input to a linear Support Vector Machine (SVM) classifier [56,57] to evaluate the discrimination capability between SCI, MCI and AD at different threshold values.

To assess the effectiveness of EpEn, we compared it to three alternative measures, commonly used in the literature: phase lag index (PLI), magnitude squared coherence (MSC), and mutual information (MI). Additionally, we considered five local graph parameters for the brain network analysis in the binary framework.

In the sequel, we describe the EEG database used and present the four connectivity measures, as well as the topological parameters used for the graph analysis.

### 2.2. Study Population

This work was carried out on a database including EEG signals of 78 subjects recorded in real clinical conditions between 2009 and 2013 at Charles-Foix Hospital (France). This retrospective study was approved by the institutional review board of the local Ethics Committee Paris 6, on 16 May 2013.

Subjects who suffered from memory impairment were steered to the outpatient memory clinic of the hospital to take several clinical and neuropsychological standard tests.

The diagnosis was established for each subject based on the clinical evaluation, brain imaging results, psychometric findings, interviews and neuropsychological tests, conducted by multidisciplinary medical staff, following the standard diagnostic criteria: DSM-IV, NINDS, Jessen criteria for SCI, McKeith criteria for Lewy body dementia [4,5,58]. We excluded from the database patients with epilepsy. It is worth noticing that the medical staff did not exploit EEG recordings in their routines to establish the diagnosis of patients in this cohort.

The study population included resting-state EEG signals of 22 SCI subjects, 28 mild to moderate AD patients and 28 MCI patients. Table 1 reports information about demographic and clinical characteristics of the patients.

EEG signals were recorded during resting-state, eyes-closed condition, using a Deltamed digital EEG acquisition system with 30 scalp electrodes placed over the head according to the 10–20 international system, as displayed in Figure 1. All signals were digitalized in a continuous recording mode for a minimum of 20 min using a sampling frequency of 256 Hz.

The EEG recordings were preprocessed off-line on MATLAB software. For each subject, continuous epochs of 20 s, free from artifacts were first selected manually after a visual inspection by experts. Then, the obtained EEG signals were band-pass filtered, using a third-order digital Butterworth filter, in the four frequency ranges: delta (1–4 Hz), theta (4–8 Hz), alpha (8–12 Hz) and beta (12–30 Hz).

### 2.3. Functional Connectivity Measures

#### 2.3.1. Epoch-Based Entropy Measure

Entropy quantifies the information content of a random variable and depends only on its probability density value. An epoch-based entropy measure (EpEn) relies on the fundamental assumption that the EEG signal can be modeled as a sequence of quasi-stationary epochs separated by abrupt transitions [20,21,22,23,24,25,26,27], as reported in Section 1.

In that sense, a Hidden Markov Model (HMM) can be considered as an appropriate statistical modeling technique to estimate the information content in piecewise stationary signals [59]. Actually, HMM can segment the signals into quasi-stationary epochs and, at the same time, perform a local estimation of the probability density on each epoch. An HMM consists of a doubly stochastic process employed to characterize the evolution of observable realizations (the captured time series), which depend on an internal process that is not directly observable, called “hidden states”. Mathematically, an HMM is defined by a finite set of states, and its transitions from one state to another are governed by “transition probabilities”. Continuous probability density function is used to characterize the relationship between states and the observable realizations. Specifically, the “emission probabilities” correspond to the conditional distributions of the observations from a given state [59].

To characterize the evolution of EEG signals over time, we naturally used a continuous left-to-right HMM structure (see Figure 2), which allows transitions from each state to itself and to its immediate right-hand neighbor only. The hidden states of the HMM correspond to the stationary segments of the signal, and the transitions correspond to the abrupt changes in the signal [59]. The EEG signal of a subject is thus represented by a succession of epochs, segmented automatically with the Viterbi algorithm using the corresponding subject’s HMM [59].

Each epoch $S_{i}$ corresponding to a hidden state of the HMM, contains a given number of observations. Each observation z in such an epoch is considered as a realization $Z_{i}$ of a random variable Z, which follows the observation probability distribution $P_{i}\left(z\right)$ modeled by a weighted sum of *M* Gaussian distributions (see Figure 2). The entropy $H^{*}\left(Z_{i}\right)$ of the epoch $S_{i}$ is computed as follows considering the ensemble of realizations of $\;Z_{i}$
(1)$$H^{*}\left(Z_{i}\right)=-\underset{z\in S_{i}}{\sum }P_{i}\left(z\right)\cdot {log}_{2}P_{i}\left(z\right)$$

Then, the entropy EpEn(Z) of the whole signal is obtained by averaging the entropy values computed for the *N* epochs
(2)$$\mathrm{EpEn}\left(Z\right)=\frac{1}{N}\overset{N}{\underset{i=1}{\sum }}H^{*}(Z_{i})$$

The use of HMM is also driven by the multichannel EEG analysis. Indeed, an HMM can manage multidimensional signals using multivariate probability density functions on such signals. To characterize the inter-relations between two EEG signals captured from two electrodes, we trained an HMM for each subject on such a coupling of EEG signals. At time *t*, a hidden state emits a two-dimensional observation vector. By applying the Viterbi algorithm, each signal is segmented into *N* epochs, and the entropy $H^{*}\left(Z_{i}\right)$ of an epoch is computed considering the probability density function estimated by the HMM on all the observations (sample points) from the two signals belonging to the associated epoch (see Figure 3). Note that although the *N* epochs are matched between EEG channels, the model does not constrain these epochs to have the same duration.

Finally, by averaging the entropy over all the *N* epochs, an epoch-based entropy value (EpEn) associated to the multivariate EEG of the subject is obtained. A high value of EpEn indicates a high information content conveyed by the coupling of two EEG signals.

#### 2.3.2. Phase Lag Index

The phase lag index (PLI) measures consistency across time of the instantaneous delay between two signals. It is largely used in the literature because of its robustness to head volume conduction, which is a common problem in EEG data [1].

PLI is computed from the asymmetry of the distribution of instantaneous signal phase differences. A non-zero phase difference (phase lag) reflects a time lag between two EEG signals [33,34]. The main approach is to neglect phase differences that are centered around 0 mod π [34]. The index of the asymmetry in the phase difference distribution is calculated as
(3)$$\mathrm{PLI}=\left|\langle sign[\sin\left(\Delta \emptyset \left(t_{k}\right)\right)\rangle \right|$$
where Δ∅  is the phase difference at time $t_{k}\;$ between two time series, computed for all sample points per epoch; sign stands for signum function; 〈.〉 indicates the mean value.

The PLI varies between 0 and 1: A zero value indicates no coupling or coupling with a phase difference centered around 0 mod π; a PLI equal to 1 indicates a perfect phase locking at a value of Δ∅. The higher this non-zero phase locking is, the higher the PLI is.

#### 2.3.3. Magnitude Square Coherence

The magnitude square coherence (MSC) measures the linear component of the functional coupling between two EEG signals *x* and *y* as a function of the frequency *f* [29,30,31,60,61,62,63]. The signals *x* and *y* are first subdivided in *M* segments of equal length *L*, then the coherence is calculated by averaging over such segments. The MSC is computed as
(4)$$c\left(f\right)=\frac{{\left|\langle X\left(f\right)Y^{*}\left(f\right)\rangle \right|}^{2}}{\left|\langle X\left(f\right)\rangle \right|\left|\langle Y\left(f\right)\rangle \right|}$$
where X(f) and Y(f) are the Fourier transforms of *x* and *y*, respectively; $Y^{*}$ is the complex conjugate of Y; |Y| is the magnitude of Y, and 〈X(f)〉 stands for the average of X(f) calculated over the *M* segments, similarly for 〈Y(f)〉 and $\langle X\left(f\right)Y^{*}\left(f\right)\rangle$.

#### 2.3.4. Mutual Information

Mutual information (MI) estimates the information gained from observations of one random variable *X* on another *Y*
(5)I(X,Y)=H(X)+H(Y)−H(X,Y)
where, H(X) and H(Y) are the Shannon entropy of *X* and *Y*, respectively, and H(X,Y) is the joint entropy of *X* and *Y*. It vanishes when *X* and *Y* are statistically independent.

On EEG signals, MI quantifies the dynamical coupling or information transmission between pairwise electrodes [28,29]. For a reliable estimation of MI, it is computed in the time–frequency domain using the normalized spectrograms as follows
(6)$$C_{x}\left(k,f\right)=\frac{{\left|X\left(k,f\right)\right|}^{2}}{\sum_{k,f}{\left|X\left(k,f\right)\right|}^{2}}$$
where the summation in the denominator is carried out over the time window *k* and frequency range *f*.

Then, the MI of the normalized spectrograms is calculated as
(7)$$I_{w}\left(C_{x},C_{y},C_{xy}\right)=\underset{k,f}{\sum }C_{xy}\left(k,f\right)log\frac{C_{xy}\left(k,f\right)}{C_{x}\left(k,f\right)C_{y}\left(k,f\right)}$$
where the normalized cross time–frequency distribution $C_{xy}\left(k,f\right)$ of *x* and *y* is computed as follows
(8)$$C_{xy}\left(k,f\right)=\frac{\left|X\left(k,f\right)Y^{*}\left(k,f\right)\right|}{\sum_{k,f}\left|X\left(k,f\right)Y^{*}\left(k,f\right)\right|}$$

### 2.4. Brain Network Analysis

In the present study, we conducted the graph analysis on binary functional connectivity matrices, by applying a thresholding approach on the generated connectivity matrices, which are originally real-valued. This allows obtaining a sparse and a binary form. Thresholding is commonly used in the literature to remove weaker connections, which are most affected by experimental noise, and to reduce the density of the graph for lower computational cost [35,36,37,38].

Absolute or proportional threshold approaches can be used. The absolute threshold approach consists of the selection of edges with a connectivity value higher than the absolute threshold value, setting all surviving connections to 1 and the others to 0 in the binary case. This leads to different density networks across subjects. To overcome this issue, we adopted in this work a proportional threshold (PT), which consists of the selection of the strongest percentage of connections in each network, setting all surviving connections to 1 and the others to 0.

In the following, we provide mathematical definitions of five commonly used and complementary local graph parameters exploited to describe network’s topology: degree, clustering coefficient, shortest path, local efficiency and betweeness.

#### 2.4.1. Degree

The degree (K) of a node reflects the importance of that node in the network. It corresponds to the number of nodes (electrodes) that still have connection with that node after thresholding. The remaining electrodes are considered as the neighbors of the node [35,36,37,38]. The degree K of a node i is defined as
(9)$${\;K}_{i}^{}=\underset{j\in N}{\sum }a_{ij}$$
where *N* is the total number of nodes in the network, and $a_{ij}$ is the connection status between *i* and *j*: $a_{ij}$ = 1 when the link between *i* and *j* exists; $a_{ij}$ = 0 otherwise.

#### 2.4.2. Clustering Coefficient

The clustering coefficient (CC) of a node estimates the density of connections established by its neighbors [37,38,40]. It is often considered as a measure of segregation: it reflects the tendency of a network to form topologically local dense circuits (cliques) presenting high strength intrinsic connectivity.

If a node i has $k_{i}$ neighbors, the clustering coefficient CC of node i is defined as
(10)$${\;\mathrm{CC}}_{i}=\frac{2\sum_{j,k}{\left(a_{ij}a_{ik}a_{jk}\right)}^{}\;}{k_{i}\left(k_{i}-1\right)}$$
where $a_{ij}$ is the connection status between nodes i and j, and $k_{i}$ is the number of connections in node i.

A high value of the local clustering coefficient indicates that the neighbors of a node i that present high strength connectivity are densely interconnected.

#### 2.4.3. Shortest Path Length

The shortest path (L) is a parameter of integration; it quantifies how the information is exchanged or integrated within the whole brain network [37,38,40]. A path is a sequence of edges that connects two nodes and its length is given by the sum of the connections forming the path.

Shortest path length (distance) $d_{i,j}^{}$ between node *i* and *j* is defined as
(11)$$d_{i,j}^{}=\underset{a_{ij}\in gi↔j}{\sum }a_{ij}$$
where gi↔j is the shortest path between nodes *i* and *j*.

The path length *L* at node *i* is defined as
(12)$$L_{i}=\frac{\sum_{i\neq j}d_{i,j}^{}}{\left(n-1\right)}$$
where *n* is the number of nodes (*n* = 30 in our study) and $d_{i,j}^{}$ is the shortest path length between nodes i and j, considering all possible paths that have to be spanned from node i to node j.

A low value of the shortest path length suggests that information is routed between electrodes with few intermediate steps (edges), which indicates rapid and high efficiency in information transmission across the network.

#### 2.4.4. Local Efficiency

The local efficiency (Eff) is another measure of segregation that performs locally at the level of the clusters retrieved with the clustering coefficient [37,38,40]. The local efficiency of the node *i* is defined as
(13)$$\;\;\;\;\;\;\;\;\;\;\;\;\;\;\;\;\;\;\;\;\;\;\;\;\;\;{\mathrm{Eff}}_{loc,i}=\frac{2\sum_{j,k}a_{ij}a_{ik}{\left(d_{j,k}^{}\right)}^{-1}}{k_{i}\left(k_{i}-1\right)}\;$$
where $d_{j,k}^{}$ is the shortest path between *j* and *k*, which contains only neighbors of *i*.

The local efficiency reports how efficient the communication is between the first neighbors *j* and *k* of the node *i* when this node is removed.

#### 2.4.5. Betweeness

The betweeness of a node (BW) is defined as the number of shortest paths in the network that pass through that node. It reflects the influence that a node has over the flow of information in a graph [37,38,40]. The betweenness of node i is defined as
(14)$${\;\mathrm{BW}}_{i}=\frac{1}{n\left(n-1\right)}\underset{h\neq j,\;h\neq i,\;j\neq i}{\sum }\frac{phj\left(i\right)}{phj}\;$$
where ρhj is the number of shortest paths between *h* and *j*, and ρhj(i) is the number of shortest paths between *h* and *j* that pass through *i*.

A node with high betweenness has a high influence on the information transmission through the network.

### 2.5. Feature Selection Method

In this study, we computed the functional connectivity measures of the three populations in the four frequency bands, and computed the associated local graph parameters for each node (*n* = 30). This induces the availability of a large number of candidate input features to the linear SVM classifier. It is, thus, necessary to perform a feature selection procedure in order to reduce the number of features by determining, upstream of the classification step, the most relevant features to discriminate between SCI, MCI and AD.

To select the most pertinent input features for the SVM classifier, we used the Orthogonal Forward Regression (OFR) algorithm and the random probe technique: the OFR algorithm allows ranking all the candidate features in decreasing order of relevance [54,55]; the random probe serves as a decision criterion to keep the most relevant features. For feature ranking with the OFR algorithm, we applied the procedure hereinafter:Select the feature that best correlates to the output of the process to be modeled. For example, in case of SCI vs. AD, the output vector contains 22 true values and 28 false values;Project the output vector onto the null space of the selected feature. Orthogonalize the remaining candidate features using Gram–Schmidt orthogonalization method;Discard the selected feature from the list of candidate features;Return to (a) and repeat the procedure until a stopping criterion is met based on the random probe technique, described below.

In order to select the most relevant features, we applied the random probe technique [53]. In the set of candidate features, we considered an additional feature, called “probe” feature, which is a realization of a random variable. This probe feature is ranked as all other candidate features by the procedure described above. It would be obvious to discard all features that are ranked after the probe. More precisely, 1000 random realizations of the probe feature are generated. Each random realization of the probe is concatenated to the set of real candidate features, and all features (real and probe) are ranked with the OFR algorithm as above-mentioned. Once the cumulative distribution of the rank of the probe was computed, we defined an acceptable risk value that a random variable can explain the output process more reliably than one of the selected real features. In this study, we chose the value 0.1 (10%) as an acceptable risk value, as carried out in [51]. Therefore, at each step of the OFR procedure, we followed this procedure:Identify a candidate feature with OFR;Compute the value of the cumulative distribution function of the rank of the probe. If the value is smaller than the risk (0.1), keep the feature and return to step (b) of the OFR algorithm; otherwise, reject the feature under consideration and finish the procedure.

## 3. Experimental Results

To discriminate between each pair of classes, i.e., SCI vs. AD, SCI vs. MCI and AD vs. MCI, for each person in the four frequency bands, we generated the connectivity matrix between all pairs of electrodes, with the four connectivity measures separate. Then, we applied on each connectivity matrix a PT value to have a binary form of such matrix (refer to Section 2.4), on which we calculated the five graph parameters.

In this study, we used 9 PT values, from 10% to 90% (with steps of 10%). Therefore, for each PT value, in each frequency band, a graph parameter vector of dimension 30 characterizes each person in the cohort.

For performance assessment with the SVM, we selected for each PT the most relevant input features to discriminate each pair of class. To do that, we first applied the feature selection algorithm (refer to Section 2.5) in each frequency band to select the most pertinent combination of electrodes that distinguish between each pair of classes. We considered an acceptable risk of 10% to fix the number of features that we had to keep using the probe method (see Section 2.5). Then, by combining all the selected features obtained on each frequency band, we applied a second feature selection to have, at the end, a combination of features fusing different frequency bands. Then, we evaluated the SVM performance by considering, progressively, the 3 most relevant features to a maximum of 10, and retained the combination of features that gave the best performance in terms of accuracy.

We report in the following sections only the performance associated to the optimal PT and the best number of relevant features, which give the best accuracy value with the linear SVM classifier.

### 3.1. SCI vs. AD

Table 2, Table 3, Table 4 and Table 5 show the performance of the SVM classifier when discriminating SCI subjects from AD patients with the five graph parameters: clustering coefficient (CC), degree (K), shortest path (L), efficiency (Eff) and betweeness (BW). Each table reports performance using a given functional connectivity measure to quantify the connectivity links in the network. 

We first observe that EpEn measure allows achieving a very good classification performance when discriminating SCI from AD. Indeed, the accuracy value is between 90% and 94% considering the five graph parameters. We reach 94% of accuracy with a good balance of specificity (proportion of well classified SCI patients) and sensitivity (proportion of well classified AD patients), considering the clustering coefficient and the efficiency.

Moreover, in the case of PLI and MSC measures, we notice that the accuracy value is 84% or 86% for three graph parameters with PLI, and for four graph parameters with MSC. In the case of MI measures, the accuracy reaches 92% with a good balance of specificity and sensitivity for almost all graph parameters, except for the shortest path for which the accuracy is 82% with a specificity of 81.82% and a sensitivity of 81.14%.

Compared to the three classical measures, we notice that EpEn allows obtaining the best performance when the brain network is characterized by the following four graph parameters: clustering coefficient, shortest path, efficiency and betweeness. Concerning the degree parameter, EpEn is ranked in the second position after MI due to a difference in sensitivity: 92.86% with MI and 89.28% with EpEn.

Finally, we can observe that, contrary to the EpEn measure, PLI, MSC and MI give, in some cases, the worst results for some graph parameters.

For a better understanding of the results of the feature selection step, the best combination of features obtained with EpEn, considering the clustering coefficient graph parameter, combines eight clustering coefficient values computed at: delta (T6), delta (CP3), delta (FP1), beta (T6), delta (FCz), beta (Pz), beta (FC3), and beta (F3). This best combination of features was obtained in the case of binary matrices binarized with a proportional threshold of 70%.

### 3.2. SCI vs. MCI

Table 6, Table 7, Table 8 and Table 9 show that the four measures allow reaching good classification performance when discriminating SCI subjects from MCI patients. This is observed especially with the MI measure that allows achieving an accuracy value between 90% and 96%, and the EpEn measure, which leads to an accuracy value between 92% and 100%. We notice that EpEn outperforms the other measures, especially for the degree, the shortest path and the betweeness parameters, reaching sometimes 100% of specificity (proportion of well classified SCI patients) and 100% of sensitivity (proportion of well classified MCI patients). The results again pointed out the reliability of the EpEn measure whatever the graph parameter used.

Additionally, we observe that PLI and MSC measures do not give the best results, whatever the graph parameter used, even though the performance is still correct.

### 3.3. AD vs. MCI

Table 10, Table 11, Table 12 and Table 13 report the classification performance of the SVM classifier when discriminating AD patients from MCI patients. Specificity and sensitivity correspond, respectively, to the proportion of AD patients and MCI patients well classified.

We first notice that all measures reach worst results compared to the other cases comparing SCI to AD or SCI to MCI. This reflects the difficulty of discriminating the AD group from the MCI group.

Moreover, we notice that PLI and MSC measures give the worst results compared to MI and EpEn. When comparing MI and EpEn, we observe that their classification performance depends on the graph parameter.

Finally, compared to the three classical measures, we notice that the EpEn measure is ranked either in the first or the second position; while the others can give the worst results for some graph parameters.

### 3.4. Global Comparison of the Four Functional Connectivity Measures

Figure 4 shows the global rank of each connectivity measure in terms of accuracy, considering all the graph parameters together and all two-class comparisons (SCI vs. AD, SCI vs. MCI and AD vs. MCI). We report the number of times each connectivity measure is ranked in position 1, 2, 3 or 4. Each connectivity measure is evaluated 15 times (5 graph parameters * 3 class pairs comparisons).

We clearly show the discriminative potential of the EpEn measure compared to the other classical measures. Indeed, EpEn is ranked ten times in the first position, four times in the second position and only one time in the third position. In addition, it was never ranked in the last position, contrary to the other connectivity measures.

We go more deeply in this comparative study by computing the average SVM posterior probability that one person is classified into the positive class for each connectivity measure and graph parameter (see Figure 5). The results show that the probability outcome for decision making with the SVM on the positive class is higher in general when considering the EpEn measure.

### 3.5. Differential AD Diagnosis with the Three Groups of Patients

This study exploits a multiclass database that includes three classes of patients: SCI, MCI and AD patients. We are, thus, left confronting a *K*-class classification problem (*K* = 3) that was turned into a set of (*K*(*K* − 1)/2) two-class problems [64], as carried out in the previous sections.

In the following, we present the results of further experiments targeting a differential AD diagnosis with the three groups of patients simultaneously, using the previous results obtained with the two-class problems.

Therefore, to assess the performance of a three-class SVM classifier, we exploited the previous linear SVM classifiers used to discriminate between each pair of classes, considering the same number *N* of selected variables. We recall that, for each PT value, we evaluated the two-class SVM classifiers by considering progressively the 3 to 10 most relevant features, and reported only the performance associated to the optimal PT and the best number of relevant features that give the best accuracy value. We follow the same methodology for the three-class problem.

To estimate pairwise posterior probabilities of the three-class SVM classifier, *N* × (*K*(*K* − 1)/2) two-class classifiers are trained for each PT (*N* = 8 since we evaluated the performance considering progressively the 3 to 10 most relevant features leading to 8 cases for each PT value).

The global probability that a patient described by the feature vector *x* belongs to the class $C_{i}$ is computed as in [64]
(15)$$\mathrm{Pr}\left(C_{i}|x\right)=1/\overset{K}{\underset{j=1,\;j\neq i}{\sum }}\frac{1}{Pr_{ij}}-\left(K-2\right)$$
where *K* is the number of classes and $Pr_{ij}$ is the probability of the patients belonging to the class *i*, estimated by the SVM classifier separating the class $C_{i}$ from the class $C_{ij}.$

Table 14, Table 15, Table 16, Table 17 and Table 18 report the best classification performance of the three-class classifier for each graph parameter using the four connectivity measures. We notice that the EpEn measure gives the best results in terms of classification performance compared to the other connectivity measures, reaching a total accuracy between 91.02% with betweenness and 94.87% with degree and efficiency parameters.

Additionally, we notice that the misclassification errors obtained with EpEn have more sense than the other measures. More precisely, most of the misclassified SCI and AD patients are classified as MCI patients, which is in accordance with the fact that MCI is an intermediate stage in the evolution towards AD.

For a better insight on the selected features, we report in Table 19, Table 20 and Table 21 the best combination of features obtained for the multiclass classification, with the EpEn measure since it gave the best performance. We can see that with the clustering coefficient, for example, the three two-classifiers (SCI vs. AD, SCI vs. MCI and AD vs. MCI) have the same number of selected features (eight features) as above-mentioned. The same is observed for the other graph parameters.

We clearly show that the selected combinations of features include different frequency bands and different electrodes. Additionally, we observe that the selected features depend on the exploited graph parameter. However, we observe a certain homogeneity between graph parameters when looking to the first selected feature, which is considered by the OFR algorithm as the best feature explaining the output. Note that the advantage of our feature selection method is to retrieve the most relevant combination of features sharing complementary information.

The electrode T6 that is located in the right side of the parieto-temporal region emerges as a relevant channel to discriminate SCI from AD with three graph parameters. Such an electrode also appears in the second position with degree parameter and in the third position with the shortest path. We also notice that the delta and theta bands are relevant to distinguish SCI from AD. For SCI vs. MCI, the first features belong in general to the posterior brain region, while for AD vs. MCI, the first features belong to the frontal brain region.

## 4. Discussion and Conclusions

In the literature, several studies on functional organization of the brain network in the context of AD reported conflicting results [41,42,43,44,45,46,48]. The observed differences among studies are mainly due to methodological differences, such as the use of different connectivity metrics to quantify the connectivity links in the brain networks. The use of different measures may reflect different processes leading to different brain network topologies. The majority of studies encouraged the use of a specific metric, without comparing it to others on the same database. Another aspect that can explain the discrepancies among studies is the exploitation of databases with different characteristics, which are sometimes prone to experimental constraints that do not match the reality on the ground.

In light of this, in the present study, we used a real-life clinical database containing EEG data of 78 patients, at SCI, MCI and mild to moderate AD stages. To our knowledge, this is the first study to date employing graph theory to study network dynamics throughout different clinical stages of cognitive decline, including healthy elders with subjective cognitive impairments (SCI), MCI patients and patients with AD.

In addition, we performed a comparison study of different functional connectivity measures for the graph theory analysis. We considered three widely used measures: phase lag index (PLI), magnitude square coherence (MSC) and mutual information (MI), relying on different mathematical concepts. We also considered the epoch-based entropy measure (EpEn), already presented in previous works [49,50,51,52] but not largely used in the literature.

The purpose of the present work was to highlight the potential use of EpEn for the detection of brain disorders with EEG signals. This measure is computed on multivariate piecewise stationary epochs using an HMM, which performs local density estimation at the epoch level. The use of an HMM is motivated by its structure, which is appropriate for modeling neural dynamics underlying the observed EEG signals. In addition, an HMM can manage multidimensional signals by estimating multivariate probability density functions on the signals.

We have proposed this statistical measure in view of what we observed in the literature. Indeed, the majority of research works share three main drawbacks. First, the reduction in EEG complexity and changes in EEG synchrony due to AD were commonly quantified separately, and only alteration in synchrony was used to quantify the connectivity links in graph networks. Second, the majority of the extracted EEG markers did not consider EEG signals as multidimensional time series. Third, such measures were computed on the whole of the EEG signals without tackling the problem of the nonstationarity of such physiological signals. The originality of our statistical entropy measure relies on the fact that it estimates the information content or the disorder in EEG signals on piecewise stationary epochs over time, and also at the spatial level, by quantifying the connectivity in terms of the heterogeneity of piecewise stationary epochs in a coupling of EEG signals. This allows a better estimation of the spatiotemporal characteristics of EEG signals merged into a single figure.

Experiments showed that the statistical modeling of EEG signals with EpEn allows a better differentiation between SCI, MCI and AD stages, compared to phase lag index, coherence and mutual information, which are deterministic measures. When discriminating SCI from AD, the accuracy value with EpEn is between 90% and 94% considering the five graph parameters. We reached 94% of accuracy with a high specificity (90.91% of well classified SCI patients) and a high sensitivity (96.43% of well classified AD patients), considering the clustering coefficient and the efficiency. Contrary to EpEn, PLI, MSC and MI, which give, in some cases, the worst results for some graph parameters.

When discriminating SCI from MCI, the results indicated that MI and EpEn lead to good classification performance: we achieved an accuracy value between 90% and 96% with MI and an accuracy value between 92% and 100% with EpEn. Nevertheless, EpEn outperforms by far the other measures, reaching sometimes 100% of specificity and sensitivity. When discriminating AD from MCI group, the results showed that PLI and MSC measures give the worst results compared to MI and EpEn. When comparing MI and EpEn, we found that their classification performance depends on the graph parameter.

After that, when summarizing all of the obtained results, we clearly showed the discriminative potential of EpEn compared to the other measures: EpEn is ranked ten times in the first position, four times in the second position, and only one time in the third position (see Figure 4). In addition, contrary to the other connectivity measures, EpEn was never ranked in the last position.

Finally, when conducting a multiclass classification to discriminate SCI, MCI and AD simultaneously, results show again that EpEn outperforms the other measures, reaching a total accuracy between 91.02% and 94.87% depending on the used graph parameter. Additionally, we noticed that the misclassification errors obtained with EpEn have more sense compared to the other measures: most of the misclassified SCI and AD patients are classified as MCI patients, which is more coherent with the evolution stages of cognitive impairment. MCI being an intermediate stage in the evolution towards AD.

In conclusion, our study demonstrates the effectiveness of the statistical modeling of EEG with an HMM for analyzing the brain network in patients with different clinical severity stages of cognitive dysfunction. However, our study presents with some limitations. We performed our experiments based on a methodology that selects automatically the most relevant input features for classification performance assessment. A deeper analysis should be performed to interpret more finely our results in terms of the selected features in relation with brain disorder detection. In addition, we reported in this study only the performance of the optimal proportional threshold value that gave the best accuracy. However, we noticed that the obtained proportional threshold value varied across the graph parameters and across the connectivity measures. It would be interesting in the future to compare all of the measures at different proportional threshold values in order to study the stability of the measures. The assumption is that some measures could be more stable to threshold changes. This could be carried out in future work.

## Figures and Tables

**Figure 1 entropy-23-01553-f001:**
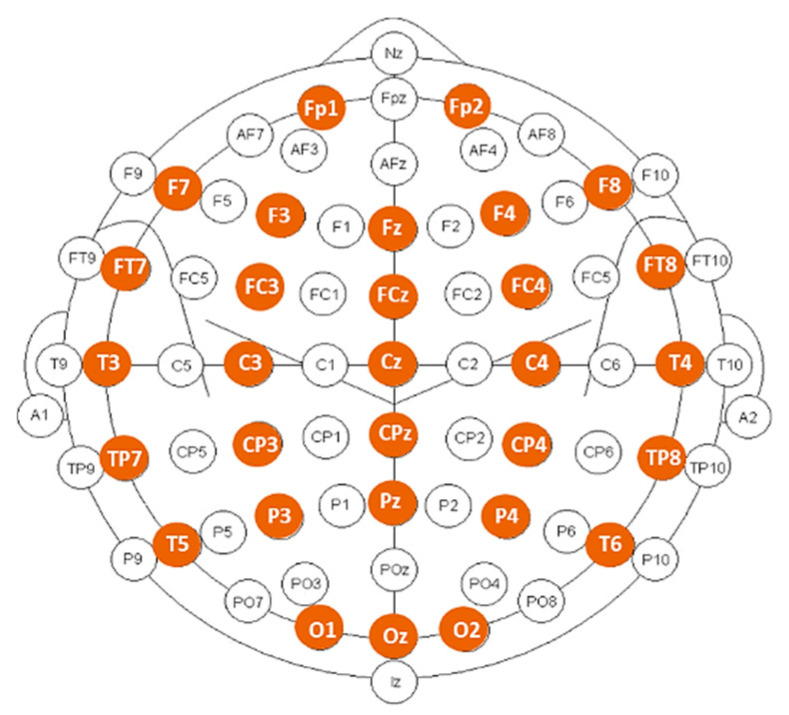
Position of the 30 electrodes used for EEG recording (marked in color).

**Figure 2 entropy-23-01553-f002:**
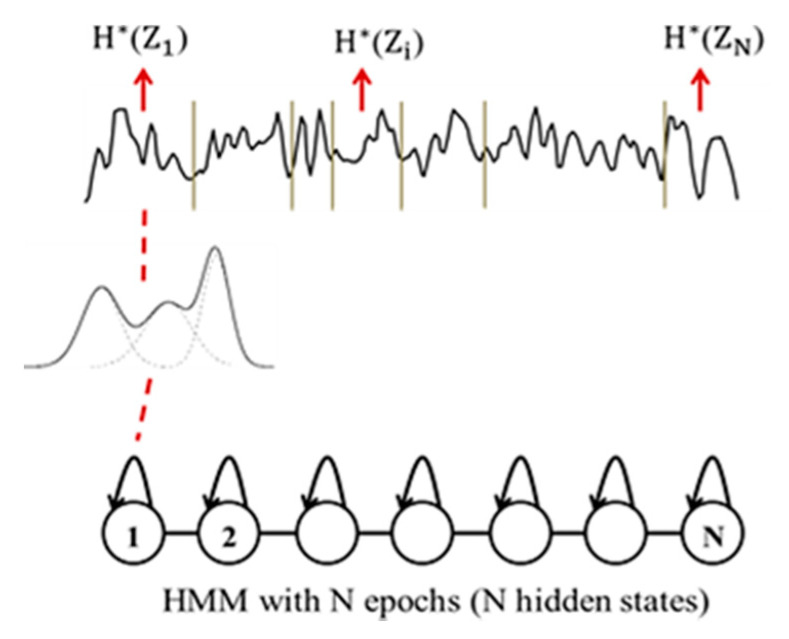
HMM modeling of an EEG signal with *N* states.

**Figure 3 entropy-23-01553-f003:**
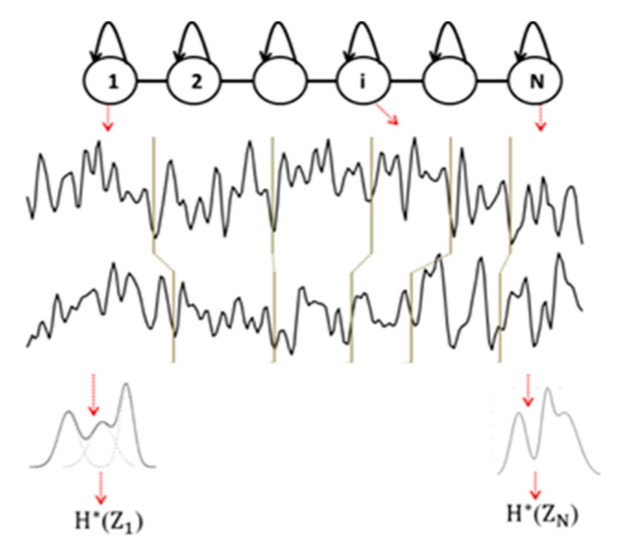
Illustration of multichannel (*D* = 2, *N* = 6) EEG signal modeling with HMM.

**Figure 4 entropy-23-01553-f004:**
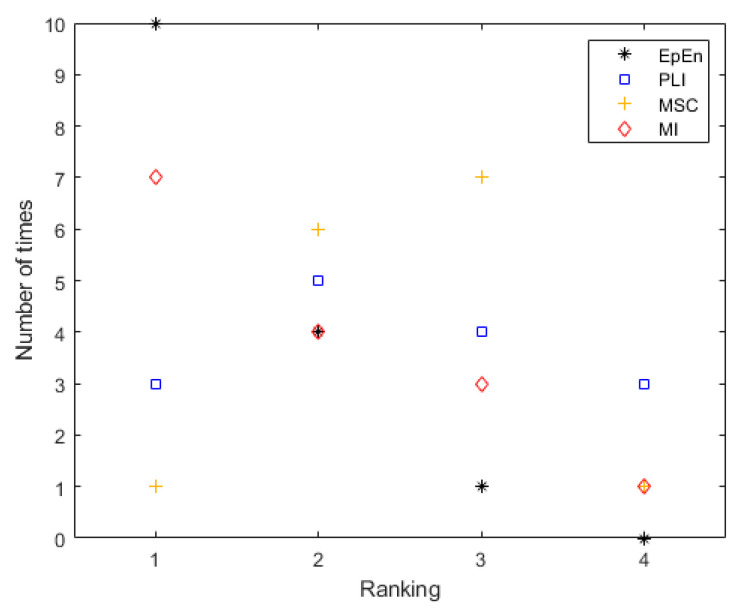
The global ranking of the four connectivity measures in terms of accuracy considering all the graph parameters and class comparisons together.

**Figure 5 entropy-23-01553-f005:**
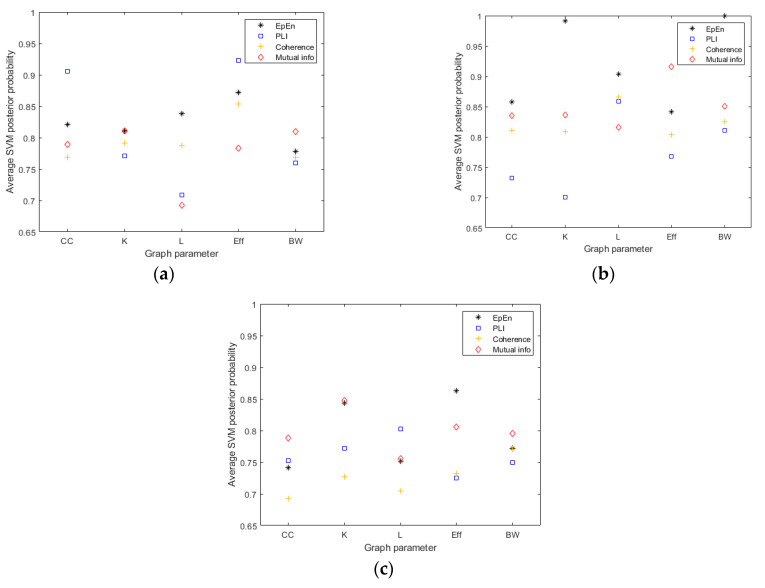
The average SVM posterior probability that one person is classified into the positive class for the four connectivity measure and the five graph parameters, when comparing: (**a**) SCI vs. AD, (**b**) SCI vs. MCI and (**c**) AD vs. MCI.

**Table 1 entropy-23-01553-t001:** Clinical characteristics of the study population. AD: Alzheimer’s disease; MCI: mild cognitive impairment; SCI: subjective cognitive impairment; MMSE: mini mental state examination; BZD: benzodiazepine.

Characteristics	SCI (*n* = 22)	MCI (*n* = 28)	AD (*n* = 28)
Age (mean ± SD)	68.9 ± 10.3	75.46 ± 9.15	80.8 ± 10.5
Female (%)	81.8%	67.8%	67.8%
MMSE (mean ± SD)	28.3 ± 1.6	23.8 ± 5.9	18.3 ± 6.1
BZD use (%)	4 (18.2%)	2 (7.1%)	8 (28.6%)
Antidepressant use (%)	2 (9.1%)	4 (14.2%)	12 (42.8%)
Neuroleptic use (%)	0	2 (7.1%)	5 (17.8%)
Hypnotic use (%)	5 (22.7%)	5 (17.19%)	7 (25%)

**Table 2 entropy-23-01553-t002:** Classification performance when discriminating SCI from AD with different graph parameters, using EpEn to quantify the connectivity links in the network.

EpEn	CC	K	L	Eff	BW
Accuracy	94%	90%	90%	94%	92%
Specificity	90.91%	90.91%	90.91%	95.45%	86.36%
Sensitivity	96.43%	89.28%	89.28%	92.86%	96.43%

**Table 3 entropy-23-01553-t003:** Classification performance when discriminating SCI from AD with different graph parameters, using PLI to quantify the connectivity links in the network.

PLI	CC	K	L	Eff	BW
Accuracy	96%	84%	84%	94%	86%
Specificity	95.45%	81.82%	77.27%	95.45%	77.27%
Sensitivity	96.43%	85.71%	89.28%	92.86%	92.86%

**Table 4 entropy-23-01553-t004:** Classification performance when discriminating SCI from AD with different graph parameters, using MSC to quantify the connectivity links in the network.

MSC	CC	K	L	Eff	BW
Accuracy	84%	86%	88%	94%	90%
Specificity	81.82%	77.27%	86.36%	90.91%	86.36%
Sensitivity	85.71%	92.85%	89.28%	96.43%	92.86%

**Table 5 entropy-23-01553-t005:** Classification performance when discriminating SCI from AD with different graph parameters, using MI to quantify the connectivity links in the network.

MI	CC	K	L	Eff	BW
Accuracy	92%	92%	82%	92%	92%
Specificity	90.91%	90.91%	81.82%	90.91%	86.36%
Sensitivity	92.86%	92.86%	82.14%	92.86%	96.43%

**Table 6 entropy-23-01553-t006:** Classification performance when discriminating SCI from MCI with different graph parameters, using EpEn to quantify the connectivity links in the network.

EpEn	CC	K	L	Eff	BW
Accuracy	94%	100%	96%	92%	100%
Specificity	90.91%	100%	95.45%	81.82%	100%
Sensitivity	96.43%	100%	96.43%	100%	100%

**Table 7 entropy-23-01553-t007:** Classification performance when discriminating SCI from MCI with different graph parameters, using PLI to quantify the connectivity links in the network.

PLI	CC	K	L	Eff	BW
Accuracy	88%	86%	94%	88%	90%
Specificity	77.27%	72.72%	86.36%	81.82%	95.45%
Sensitivity	96.43%	96.43%	100%	92.86%	85.71%

**Table 8 entropy-23-01553-t008:** Classification performance when discriminating SCI from MCI with different graph parameters, using MSC to quantify the connectivity links in the network.

MSC	CC	K	L	Eff	BW
Accuracy	90%	88%	94%	94%	90%
Specificity	81.82%	81.82%	90.91%	90.91%	86.36%
Sensitivity	96.43%	92.85%	96.43%	96.43%	92.86%

**Table 9 entropy-23-01553-t009:** Classification performance when discriminating SCI from MCI with different graph parameters, using MI to quantify the connectivity links in the network.

MI	CC	K	L	Eff	BW
Accuracy	94%	90%	90%	96%	92%
Specificity	95.45%	86.36%	90.91%	90.91%	86.36%
Sensitivity	92.86%	92.86%	89.28%	100%	96.43%

**Table 10 entropy-23-01553-t010:** Classification performance when discriminating AD from MCI with different graph parameters, using EpEn to quantify the connectivity links in the network.

EpEn	CC	K	L	Eff	BW
Accuracy	87.5%	89.28%	83.93%	91.07%	89.28%
Specificity	89.28%	92.86%	89.28%	89.28%	85.71%
Sensitivity	85.71%	85.71%	78.57%	92.86%	92.86%

**Table 11 entropy-23-01553-t011:** Classification performance when discriminating AD from MCI with different graph parameters, using PLI to quantify the connectivity links in the network.

PLI	CC	K	L	Eff	BW
Accuracy	87.5%	89.28%	83.93%	85.71%	87.5%
Specificity	85.71%	89.28%	89.28%	89.28%	85.71%
Sensitivity	89.28%	89.28%	78.51%	82.15%	89.28%

**Table 12 entropy-23-01553-t012:** Classification performance when discriminating AD from MCI with different graph parameters, using MSC to quantify the connectivity links in the network.

MSC	CC	K	L	Eff	BW
Accuracy	80.36%	83.93%	80.36%	83.93%	87.5%
Specificity	71.43%	89.28%	75%	82.14%	89.28%
Sensitivity	89.28%	78.57%	85.71%	85.71%	85.71%

**Table 13 entropy-23-01553-t013:** Classification performance when discriminating AD from MCI with different graph parameters, using MI to quantify the connectivity links in the network.

MI	CC	K	L	Eff	BW
Accuracy	85.71%	92.86%	89.28%	91.07%	85.71%
Specificity	82.14%	92.86%	92.86%	92.86%	85.71%
Sensitivity	89.28%	92.86%	85.71%	89.28%	85.71%

**Table 14 entropy-23-01553-t014:** Confusion matrices for differential AD diagnosis with the three groups of patients, using the clustering coefficient (CC) parameter with the four connectivity measures.

CC		SCI	MCI	AD	Total Accuracy
**EpEn**	SCI (*n* = 22)	20	2	0	92.31%
	MCI (*n* = 28)	0	25	3	
	AD (*n* = 28)	0	1	27	
**PLI**	SCI (*n* = 22)	21	1	0	91.03%
	MCI (*n* = 28)	0	26	2	
	AD (*n* = 28)	1	3	24	
**MSC**	SCI (*n* = 22)	19	2	1	87.18%
	MCI (*n* = 28)	3	24	1	
	AD (*n* = 28)	0	4	24	
**MI**	SCI (*n* = 22)	19	0	3	89.74%
	MCI (*n* = 28)	0	26	2	
	AD (*n* = 28)	1	3	24	

**Table 15 entropy-23-01553-t015:** Confusion matrices for differential AD diagnosis with the three groups of patients, using the degree (K) parameter with the four connectivity measures.

K		SCI	MCI	AD	Total Accuracy
**EpEn**	SCI (*n* = 22)	21	0	1	94.87%
	MCI (*n* = 28)	0	26	2	
	AD (*n* = 28)	1	0	27	
**PLI**	SCI (*n* = 22)	17	1	4	88.46%
	MCI (*n* = 28)	0	25	3	
	AD (*n* = 28)	1	0	27	
**MSC**	SCI (*n* = 22)	20	2	0	89.74%
	MCI (*n* = 28)	2	23	3	
	AD (*n* = 28)	0	2	26	
**MI**	SCI (*n* = 22)	20	1	1	89.74%
	MCI (*n* = 28)	5	22	1	
	AD (*n* = 28)	0	1	27	

**Table 16 entropy-23-01553-t016:** Confusion matrices for differential AD diagnosis with the three groups of patients, using the shortest path (L) parameter with the four connectivity measures.

L		SCI	MCI	AD	Total Accuracy
**EpEn**	SCI (*n* = 22)	20	2	0	91.03%
	MCI (*n* = 28)	2	24	2	
	AD (*n* = 28)	1	1	26	
**PLI**	SCI (*n* = 22)	19	2	1	85.90%
	MCI (*n* = 28)	0	25	3	
	AD (*n* = 28)	2	3	23	
**MSC**	SCI (*n* = 22)	20	1	1	87.18%
	MCI (*n* = 28)	2	23	3	
	AD (*n* = 28)	1	3	24	
**MI**	SCI (*n* = 22)	20	0	2	91.03%
	MCI (*n* = 28)	3	23	2	
	AD (*n* = 28)	0	1	27	

**Table 17 entropy-23-01553-t017:** Confusion matrices for differential AD diagnosis with the three groups of patients, using the efficiency (Eff) parameter with the four connectivity measures.

Eff		SCI	MCI	AD	Total Accuracy
**EpEn**	SCI (*n* = 22)	21	1	0	94.87%
	MCI (*n* = 28)	0	27	1	
	AD (*n* = 28)	0	2	26	
**PLI**	SCI (*n* = 22)	16	5	1	89.74%
	MCI (*n* = 28)	2	25	1	
	AD (*n* = 28)	0	0	28	
**MSC**	SCI (*n* = 22)	17	5	0	89.74%
	MCI (*n* = 28)	2	25	1	
	AD (*n* = 28)	0	1	27	
**MI**	SCI (*n* = 22)	19	1	2	89.74%
	MCI (*n* = 28)	2	25	1	
	AD (*n* = 28)	1	2	25	

**Table 18 entropy-23-01553-t018:** Confusion matrices for differential AD diagnosis with the three groups of patients, using the betweeness (BW) parameter with the four connectivity measures.

BW		SCI	MCI	AD	Total Accuracy
**EpEn**	SCI (*n* = 22)	19	2	1	91.02%
	MCI (*n* = 28)	0	28	0	
	AD (*n* = 28)	1	3	24	
**PLI**	SCI (*n* = 22)	17	1	4	85.90%
	MCI (*n* = 28)	2	24	2	
	AD (*n* = 28)	3	0	25	
**MSC**	SCI (*n* = 22)	19	2	1	91.03%
	MCI (*n* = 28)	0	28	0	
	AD (*n* = 28)	3	1	24	
**MI**	SCI (*n* = 22)	18	1	3	89.74%
	MCI (*n* = 28)	0	27	1	
	AD (*n* = 28)	1	2	25	

**Table 19 entropy-23-01553-t019:** Best combination of features for discriminating SCI from AD patients using different graph parameters with EpEn.

Selected Features for SCI vs. AD										
CC	θ_T6	β_P4	α_P3	θ_F4	δ_F8	δ_FC4	δ_F4	δ_T5		
K	α_FC4	δ_T6	β_CPz	β_Fp2	δ_P3	δ_C3	α_TP8	θ_TP8	δ_FC3	
L	δ_F3	α_Fp2	δ_T6	δ_P4	θ_Oz	α_FC4	β_FC4	β_Fp2	β_Oz	
Eff	δ_T6	δ_F3	α_CP3	δ_FT8	θ_TP7	α_FT8	α_Fz	β_FC4	β_FC3	β_P4
BW	δ_T6	δ_P3	β_FC4	β_FC3	β_FCz	δ_C4	β_T6	θ_T5		

**Table 20 entropy-23-01553-t020:** Best combination of features for discriminating SCI from MCI AD patients using different graph parameters with EpEn.

Selected Features for SCI vs. MCI										
CC	α_P3	δ_Fp1	δ_CPz	θ_T6	α_C3	θ_TP7	δ_F3	α_TP8		
K	β_O2	θ_F8	α_FC4	δ_T6	δ_F8	δ_T3	δ_Oz	θ_T4	δ_FC4	
L	δ_Oz	β_Oz	β_Fp2	β_C4	β_CP3	β_F8	β_T3	β_Pz	α_FC4	
Eff	δ_F3	β_TP7	θ_FT7	α_P3	θ_TP7	β_FC4	β_O1	α_Fz	α_FT8	α_C4
BW	β_T6	δ_FC3	δ_Fz	α_F8	δ_FT8	β_F4	β_O2	β_Fz		

**Table 21 entropy-23-01553-t021:** Best combination of features for discriminating AD from MCI patients using different graph parameters with EpEn.

Selected Features for AD vs. MCI										
CC	δ_Fp1	δ_T3	δ_FC4	δ_F4	β_P3	θ_Fz	θ_FT8	α_Fp2		
K	δ_F3	θ_Fz	δ_Fz	δ_Cz	δ_F4	δ_FC3	δ_FT8	β_T4	θ_TP7	
L	β_F8	δ_Fp1	β_O2	δ_T3	θ_Cz	β_CP4	β_T4	θ_FT8	δ_P3	
Eff	δ_Fp1	δ_T3	α_T3	θ_Fz	α_Fp1	β_C4	β_Cz	δ_CPz	α_F8	β_F7
BW	β_Fp1	β_Fz	θ_CPz	β_P4	β_TP7	δ_Fz	δ_T6	β_O2		

## Data Availability

We conducted a retrospective study on EEG data of patients referred to the outpatient memory clinic of the Charles-Foix hospital (France). This retrospective study was approved by the institutional review board of the local Ethics Committee Paris 6, on 16 May 2013 (France). All data were fully anonymized before exploiting them in our research work. Before conducting this retrospective study, an information letter on the research study was sent to the patients, with the possibility of them opposing the use of their collected data. Informed consent was thus waived in this context. Data are available upon request to researchers qualified to handle confidential data. Data requests can be made to Laurent Capelle (email: cppidf6.salpetriere@yahoo.fr, Comité de protection des personnes Ile-de-France VI) or from the corresponding author (kiyoka.kinugawa@aphp.fr) upon reasonable request. There are legal and ethical restrictions on sharing these data. The French law requires patients to be duly informed of any use of their data. We did not obtain patients’ permission to share their data publicly. We are not able to ask for their consent today because the data was collected some time ago, between 2009 and 2013.

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
