# Peer review of "A Comparative Study of Functional Connectivity Measures for Brain Network Analysis in the Context of AD Detection with EEG"

_entropy, 2021, doi:10.3390/e23111553_

Round 1

Reviewer 1 Report

Summary:

In this study, the authors set to verify the utility of epoch-based entropy (EpEn) in comparison to mutual information, coherence, and phase lag index for pair-wise classification (using SVM classifier) of EEG data of subjective cognitive impairment (SCI) patients, mild cognitive impairment (MCI) patients, and AD patients.

Overall, the manuscript is well-written and is easy to read. Although, it would benefit from further auditing and proofread.

However, there are a few points that require the authors’ further clarification. They are as follows.

Major Comments:

Lines 140-141: “… to discriminate between each pair of classes, i.e. SCI vs. AD , SCI vs. MCI, and AD vs. MCI.” Why would the authors consider a paired classification? Why did they not simply perform a three-class classification. This is a normal procedure and more reliable to realize the specificity and discriminatory power of each measure examined.

Line 186: “Epoch-based entropy measure (EpEn) relies on the fundamental assumption that the EEG signal is piecewise stationary; it can be viewed as being stationary at the time scale 187 of an epoch.”: One of main drawback that the authors emphasize on in the Introduction is assumption of stationarity of EEG signal by other measures and that such an assumption is not warranted. How would then the authors justify this, using the authors terminology, “fundamental assumption” by EpEn? This is particularly pressing since epochs, by definition, would have shorter time series than the actual signal and therefore are more susceptible to non-stationarity?

In the same vein,

Lines 190-191: “Actually, HMM can segment the signals into quasi-stationary epochs, and at the same time perform a local estimation of the probability density on each epoch.” What do the authors mean by this sentence?

Line 230: “.. associated to the multivariate EEG of the subject is computed.”: Does this mean that the authors average over all pair-wise EpEn values for a subject, thereby obtaining a single (i.e., grand average / global EpEn), per subject?

Lines 312-314: “In the following, we provide mathematical definitions of five commonly used and complementary graph parameters: degree, clustering coefficient, shortest path, efficiency 313 and betweeness.” All these measures can be computed for real-valued graphs and networks. As such, it is unclear why the authors opted for binarization of their networks/graphs?

Also, while using the threshold for binarization; although using mean (or better, median that is unaffected by outliers) is quite standard, a more systematic/discipline approach should also consider the significance of the calculated values. Specifically, the authors could compute the confidence interval of their computed values, given a proper p-value (e.g., in the present case, 0.05/3 given that there are three classes). They can then consider the upper-bound of their confidence interval as threshold.

This procedure could also be considered for feature selection whose threshold appears to be arbitrarily chosen (Line 413: “Then, we defined an acceptable risk (10%)…”)

Lines 614-615: “Experiments showed that the statistical modeling of EEG signals with EpEn allows a better differentiation between SCI, MCI and AD stages, compared to Phase-Lag index, Coherence and Mutual Information, which are deterministic measures.”: What do the authors mean by “deterministic”?

The data preprocessing steps requires further explanation. How did they remove e.g., low-frequency drifts, eye-movement artifacts, heartbeat, and respiration? Did they apply average-referencing of their signal?

The authors also appear not to consider the gamma frequency band. What was the rationale behind this decision? Gamma is involved in many higher-order cognitive functions.

Minor Comments:

Figures 2 & 3: the arrows on between-state transitions appear to be missing.

Author Response

 Answer to reviewers

We sincerely thank the reviewers for their interesting feedback, which helped us to improve our paper, revised according to their remarks and suggestions. We have addressed the reviewers’ comments and have modified our manuscript accordingly.

In general, following the remarks of Reviewers:

  • We improved the introduction by clarifying the non-stationarity nature of EEG signals. We have added references that address this issue.
  • We added in “Material and Methods” a Section 2.1 “Methodology”, which presents the adopted methodology to conduct the comparative study of connectivity measures.
  • We clarified in Section 2.3.1 the use of HMM for EEG signal modeling and re-wrote some parts to describe the computation of EpEn.
  • We improved the description of the feature selection method in Section 2.5 for a better understanding.
  • We added a three-class classification in Section 3.5, as asked by Reviewer 1.
  • We also added a short analysis on the obtained selected features at the end of Section 3.5.
  • We also improved the visualization of Figures 2, 3, 4 and 5.

Besides, we rephrase some parts in the manuscript as asked by the Editor, and made some writing improvements to make the paper more clear, especially concerning the description of the methodology.

Also, we corrected some statistics in Table 1. In addition, we corrected some parts in Section 2.4 on graph parameters to be more appropriate to the binary graph analysis framework.

In the attached PDF file, our responses to the two reviewers point by point.

Here after our responses to the two reviewers point by point.

Reviewer 2 Report

In the present work the authors compare the predictive efficacy of four measures of functional connectivity (Epoch-based entropy, Phase-Lag index, Coherence, and Mutual Information), in three groups of patients (subjective cognitive impairment patients, mild cognitive impairment patients, and Alzheimer’s disease patients), applied to resting-state EEG recordings.

The four functional connectivity measures are computed between all pairs of electrodes for each participant in four frequency bands. After a binarization process with different thresholds, five graph-theory measures were obtained from the connectivity matrices for each frequency band: degree, clustering coefficient, shortest path, efficiency and betweeness. Next, the SVM classifier was used to filter the most relevant characteristics with respect to the discrimination capacity between groups.

The results obtained indicate that, although all the measures studied allow discrimination between groups, the epoch-based entropy method is the best analyzing the brain network in patients with different clinical severity stages of cognitive dysfunction.

I believe that it is a well-written paper, and that, in general, it presents clear results regarding the classification effectiveness of the method that the authors propose (Epoch-based entropy). However, reading this paper has raised some questions for me, and I would like to know the opinion of the authors before making a final decision on this work.

First, at the methodological level, the authors have made several decisions. For example, using SVM to select features, instead of other techniques such as Bayes or KNN, or for example, to compare the EpEnt, with Epoch-based entropy, Phase-Lag index, Coherence, and Mutual Information and not others such as transfer entropy, granger or correlation, or for example, calculating degree, clustering coefficient, shortest path, efficiency and betweenness instead of others such as cost, diameter or participation coefficient. It might be interesting to justify these decisions as this could help other researchers who will do similar work in the future.

Regarding the possible implications for readers with a more clinical profile, the paper is a bit opaque in relation to the features that work best in discriminating between groups of patients. In this sense, it would be interesting to know which electrodes or which frequency bands performed best, or whether these features vary or not between the pairs of groups being compared.

Regarding the sample, it appears that the authors in this paper again use the same database they already used in Houmani N, Vialatte F, Gallego-Jutglà E, Dreyfus G, Nguyen-Michel V-H, Mariani J, et al. (2018) Diagnosis of Alzheimer's disease with Electroencephalography in a differential framework. PLoS ONE 13(3): e0193607. https://doi.org/10.1371/journal.pone.0193607. However, in the present paper, although the SCI and MCI groups appear to be the same, the AD patient group is of a different size than the 2018 paper. A careful reading of the sample description makes you realize that here the authors work only with "mild to moderate AD patients", and this could explain the different sample size. However, a justification for this decision would be welcome.

Finally, the groups being compared appear to differ quite a bit in variables that could affect EEG recordings. For example, age or medication are factors that we know affect EEG signals. It would be interesting to know the authors' opinion in this regard.

As minor aspects that could be taken into consideration, in Figure 4 it would be good to use the same units on the x-axis for all histograms. I also think that Figure 5 could be improved. The x-axis categories could all be in the same font. In addition, I am not sure that a line graph is the most appropriate to describe these factors.

Author Response

 Answer to reviewers

We sincerely thank the reviewers for their interesting feedback, which helped us to improve our paper, revised according to their remarks and suggestions. We have addressed the reviewers’ comments and have modified our manuscript accordingly.

In general, following the remarks of Reviewers:

  • We improved the introduction by clarifying the non-stationarity nature of EEG signals. We have added references that address this issue.
  • We added in “Material and Methods” a Section 2.1 “Methodology”, which presents the adopted methodology to conduct the comparative study of connectivity measures.
  • We clarified in Section 2.3.1 the use of HMM for EEG signal modeling and re-wrote some parts to describe the computation of EpEn.
  • We improved the description of the feature selection method in Section 2.5 for a better understanding.
  • We added a three-class classification in Section 3.5, as asked by Reviewer 1.
  • We also added a short analysis on the obtained selected features at the end of Section 3.5.
  • We also improved the visualization of Figures 2, 3, 4 and 5.

Besides, we rephrase some parts in the manuscript as asked by the Editor, and made some writing improvements to make the paper more clear, especially concerning the description of the methodology.

Also, we corrected some statistics in Table 1. In addition, we corrected some parts in Section 2.4 on graph parameters to be more appropriate to the binary graph analysis framework.

In the attached PDF file, our responses to the two reviewers point by point.

Round 2

Reviewer 1 Report

Thank you for addressing my comments and also extending the content of the manuscript to provide further explanation and results. The manuscript is substantially improved and is, in this reviewer's opinion, ready for publication.